Extended Abstract Track

# Learning and Shaping Manifold Attractors for Computation in Gated Neural ODEs

**Timothy Doyeon Kim**                                    TDKIM@PRINCETON.EDU
*Princeton Neuroscience Institute (PNI)*
*Princeton University, Princeton, NJ*

**Tankut Can**\*                                          TANKUT@IAS.EDU
*School of Natural Sciences*
*Institute for Advanced Study, Princeton, NJ*

**Kamesh Krishnamurthy**\*                               KAMESHK@PRINCETON.EDU
*Joseph Henry Laboratories of Physics and PNI*
*Princeton University, Princeton, NJ*

**Editors:** Sophia Sanborn, Christian Shewmake, Simone Azeglio, Arianna Di Bernardo, Nina Miolane

## Abstract

Understanding how the dynamics in biological and artificial neural networks implement the computations required for a task is a salient open question in machine learning and neuroscience. A particularly fruitful paradigm is computation via dynamical attractors, which is particularly relevant for computations requiring complex memory storage of continuous variables. We explore the interplay of attractor geometry and task structure in recurrent neural networks. Furthermore, we are interested in finding low-dimensional effective representations which enhance interpretability. To this end, we introduce gated neural ODEs (gnODEs) and probe their performance on a continuous memory task. The gnODEs combine the expressive power of neural ordinary differential equations (nODEs) with the trainability conferred by gating interactions. We also discover that an emergent property of the gating interaction is an **inductive bias** for learning (approximate) continuous (manifold) attractor solutions, necessary to solve the continuous memory task. Finally, we show how reduced-dimensional gnODEs retain their modeling power while greatly improving interpretability, even allowing explicit visualization of the manifold attractor geometry.

**TL;DR:** Gated neural ODEs effectively learn interpretable, low-dimensional manifold geometries to solve continuous memory tasks.

**Keywords:** Computational Neuroscience, Continuous Attractor Geometry, Dynamical Systems, Differential Equations, Neural ODEs, Gating, Interpretability

## 1. Introduction

How do parts of a neural computation map on to the dynamical motifs exhibited by the network? Recurrent neural networks (RNNs) are the model of choice for both implementing tasks which are dynamical in nature (Jozefowicz et al., 2015) and understanding the link between dynamics and computation (Vyas et al., 2020). In this work, we are particularly interested in tasks which involve storing or integrating a continuous signal. Continuous attractors (CAs) are a natural candidate for providing the dynamical substrate for integrators, by utilizing a manifold of fixed points to continuously manipulate and read out memories.

---

\* These co-corresponding authors contributed equally to this work.

# Extended Abstract Track

However, most models of CAs are highly fine-tuned, and consequently not robust to perturbations. This raises some doubt as to the ability of a simple gradient-based learning algorithm to find such fine-tuned attractors. We extend previous work analyzing the ability of RNNs to learn CAs (see e.g., Maheswaranathan et al. (2019)), providing a partial explanation of these findings and a novel architecture which allows one to visualize and interpret the geometry of the CA representations.

The form of RNNs often considered in the neuroscience, physics and cognitive-science literature assumes that the interaction between units is additive (McCulloch and Pitts, 1943; Sompolinsky et al., 1988; Elman, 1990; Vogels et al., 2005; Sussillo and Abbott, 2009; Song et al., 2016; Yang et al., 2019); however, gating or multiplicative interactions (Hochreiter and Schmidhuber, 1997; Cho et al., 2014) may be incorporated into these RNNs to train them on tasks involving long timescales. In our work, we observe that gating interactions give rise to an emergent **inductive bias** for learning approximate continuous attractors.

This still leaves the challenge of interpreting the (usually) high-dimensional representations required to learn such attractors. On this note, we may turn to neural ODEs (nODEs), a class of ODE models with a velocity field parametrized by a deep neural network (DNN), which can potentially implement more complex computations **in lower dimensions** than classical RNNs (Chen et al., 2018; Kidger, 2022). This increased complexity in lower latent/phase-space dimensions subsequently helps in extracting interpretable, *effective* low-dimensional dynamics that may underlie a dataset or task (Kim et al., 2021).

In this work, we explore the ability of RNNs and nODEs, gated and not, to learn continuous attractors shaped by a synthetic variable-amplitude flip-flop task in Section 2. Our main finding is that networks with gating interactions tend to strongly favor learning marginally-stable fixed points, leading us to the conclusion that gating gives rise to an inductive bias for continuous attractors. We further demonstrate the **interpretability** of the gnODEs' solutions, found via stochastic gradient descent, for which we can easily visualize the geometry of the continuous attractor and its influence on generalization.

The **gated neural ODE** (gnODE) is the most general model we study, and described by

$$\tau \dot{\mathbf{h}} = \sigma\left(W_z \mathbf{h} + U_z \mathbf{x} + \mathbf{b}_z\right) \odot \left[-\mathbf{h} + F_\theta(\mathbf{h}, \mathbf{x})\right]. \tag{1}$$

Here, $\mathbf{h} \in \mathbb{R}^N$ is the hidden/internal state vector, $N$ is the phase-space (or latent) dimension, $\mathbf{x}(t) \in \mathbb{R}^D$ is the input vector, $\tau$ is the time constant, with the sigmoidal gate $\sigma(x) = [1 + e^{-x}]^{-1}$, and the velocity vector field $F_\theta : \mathbb{R}^N \times \mathbb{R}^D \to \mathbb{R}^N$ parameterized by a DNN (see Appendix A for details).

Without the gating interaction (i.e., setting $\sigma = 1$), this reverts to a form in which nODEs are typically studied (Chen et al., 2018). With the choice $F_\theta(\mathbf{h}, \mathbf{x}) = \phi(W\mathbf{h} + U\mathbf{x}(t) + \mathbf{b})$, (1) reduces to a "minimal gated recurrent unit" (mGRU; Ravanelli et al. (2018)), which is a simplified version of the popularly used gated recurrent unit (GRU; Cho et al. (2014)). When in addition the gating interaction is removed ($\sigma = 1$), (1) reduces to a widely studied class of models known as "Elman" (or "vanilla") RNNs. The mGRU was shown in Can and Krishnamurthy (2021); Krishnamurthy et al. (2022) to exhibit a manifold of marginally-stable fixed points in the limit of step-like gating function $\sigma$ for a wide range of parameters. This property is likely involved in shaping the inductive bias of gated nets, since it is useful in tasks requiring memory of continuous quantities.

We discuss related work on nODEs and gating in Appendix B.

## 2. Variable-Amplitude Flip-Flop Task

We consider a modification of the classic "3-bit flip-flop task" studied in Sussillo and Barak (2013), in which the network is given a continuous stream of inputs coming from 3 independent channels. In each channel, a transient pulse of value either $+1$ or $-1$ is emitted at random times, and the network output must maintain the value of the most recent pulse in each channel (see Appendix D for an illustration, and the results of training gnODE and other gated networks on this task).

We modify this task by allowing for real-valued pulses sampled uniformly from an interval (e.g., $[-1, 1]$) (Figure 1A). When we set the phase-space dimension of our networks to be $N = 6$, we find that gnODE successfully reached validation mean-squared error (MSE) $< 0.01$ with appropriate hyperparameters, while for other networks, all runs had MSEs $\geq 0.025$ (Figure 1B top). We verified that the validation MSEs of our networks converged after training (Figure 1B bottom). This suggests that **only gnODE is able to solve the task accurately when the phase-space dimension is low.**

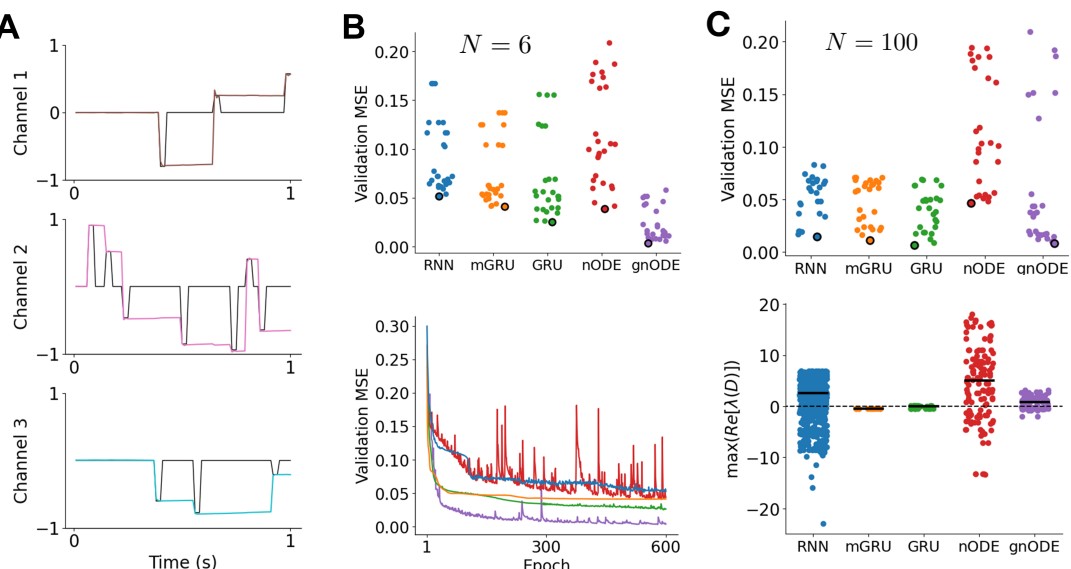

Figure 1: (A) An example validation trial of the variable-amplitude 3-bit flip-flop task with inputs in each channel shown in black, and the trained gnODE traces maintaining the previous pulse value shown in colors. (B) **top** Each circle represents the minimum validation MSE achieved during 600 epochs of training for a particular set of hyperparameters. Circles with black edges represent the minimum out of this set; **bottom** Validation loss traces as a function of epoch is shown for the circles with black edges, with color codes same as above. (C) **top** same as (B) **top** but for $N = 100$; **bottom** The distribution of spectral abscissa at fixed points.

One reason why the gnODE performs much better than the the vanilla RNN, mGRU and GRU is that a gnODE can achieve overparametrization (which may be critical for

performance) without increasing its phase-space dimensionality, while for a typical vanilla RNN, mGRU or GRU, we would need to increase their dimensionality for more parameters. We do find that when the phase-space dimension is high (i.e., $N = 100$), the vanilla RNN, mGRU and GRU can solve the task (Figure 1C). However, when we applied PCA to the trajectories taken by these high-dimensional networks, we needed at least 10 principal components to explain 0.9 of the variance, suggesting that the high-dimensional networks do not necessarily favor low-dimensional solutions in this setup, even though, a gnODE with $N = 3$ can easily solve this task (MSE < 0.01).

**Fixed-points and Marginal Stability:** Following Sussillo and Barak (2013), to further examine how these networks solve the task, we use Newton's method to find approximate fixed points (see Appendix D.3 for details). For each network that solved the task, we computed the maximum real component of the eigenvalues (i.e., **spectral abscissa**) of the numerical Jacobian obtained from each of the detected fixed points. The distribution of these spectral abscissa show that the medians and the quartiles of the gated networks (mGRU, GRU, gnODE) are closer to zero, compared to those of the vanilla networks (vanilla RNN, nODE) (Figure 1C bottom; Table 1). Together, these (Figure 1C top and bottom) suggest that the vanilla RNN utilizes a combination of marginally-stable and stable fixed points, while the gated networks mostly rely on marginally-stable fixed points to reach their solutions. We obtained similar results for networks assuming $N = 6$, although for these networks, only gnODE reached validation MSE < 0.01 (see Appendix D.4 for details).

**Geometry and Interpretability of Solutions:** For ease of visualization, we trained a 2-dimensional gnODE and its variants on the 2-bit flip-flop task and plot the 2-d flow field projected onto the Channel 1 and Channel 2 output axes. For the fixed-amplitude task (where the pulse values can either be +1 or −1), we find 4 stable fixed points, and find that each input perturbation moves the gnODE state from exactly one stable fixed point to another (Figure 2A). We then trained a gnODE on a variable-amplitude 2-bit flip-flop task where the pulses can take real values from −1 to +1. When we plot the flow field in the output space, we see that the velocity of the flows are close to zero, and this plane of fixed points roughly form a square between −1 and +1. Input perturbations try to move the gnODE state within the square, so that gnODE can hold onto the memory of the inputs (Figure 2B). In summary, the gnODE learns a **continuous attractor** in the shape of a square, and is solving the variable-amplitude flip-flop task in an intuitively appealing way, by simply integrating the input. We also note that no other network was able to solve this task in such low phase-space dimension, suggesting gnODEs are best suited for studying the emergence of *interpretable* representations used to solve this task.

The plane of fixed points show up not only for this particular task but also for other tasks. When we change the statistics of the pulses, we find attractors with different geometries, e.g., a **rectangular attractor** (Figure 2C) or a **disk attractor** (Figure 2D). We find that when the input pulses fall inside the continuous attractor, gnODE is able to generalize even if it has not seen the pulses during training, while gnODE is not able to generalize to pulses that fall outside the continous attractor (see Figures in Appendix D.5 for flow fields of gnODE trained on more variants of the 2-bit flip-flop task and more discussion on how the geometry of attractor solutions found by the gnODE is related to its generalization capacities).

Extended Abstract Track

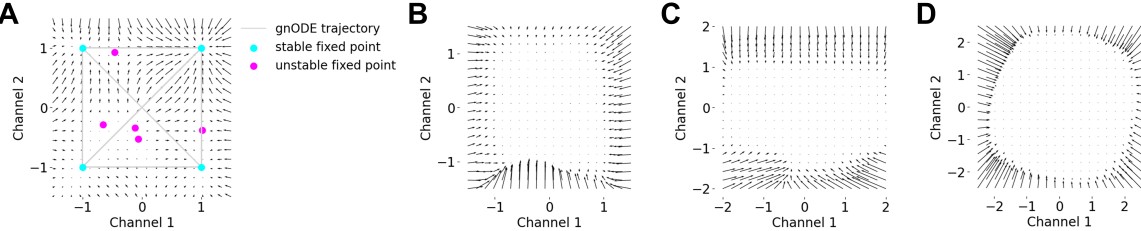

Figure 2: Flow fields of gnODE with $N = 2$ performing four different versions of the 2-bit flip-flop task. The input pulses $C_1$ in Channel 1 and $C_2$ in Channel 2 are given by (A) $C_1, C_2 \in \{-1, +1\}$. The light gray lines indicate example trajectories taken by gnODE. (B) **square**: real-valued input pulses in the interval $C_1, C_2 \in [-1, 1]$. (C) **rectangle**: $C_1 \in [-2, 2]$ and $C_2 \in [-1, 1]$. (D) **annulus**: $1 \leq \sqrt{C_1^2 + C_2^2} \leq 2$.

## 3. Conclusion

We introduced gated neural ordinary differential equations (gnODEs), a novel nODE architecture which utilizes a gating interaction to dynamically modulate the timescale.

We used the variable-amplitude $n$-bit flip-flop task as an example that requires robust memory capabilities to demonstrate the inductive bias of the gnODEs to learn continuous attractors. We also showed that, compared to other architectures, the gnODE can learn this task with a lower phase-space dimension. When we changed the statistics of the pulses, we found that the gnODE is capable of learning the shape of the attractor that reflects such change, and that the gnODE generalizes to inputs that are not seen during training only if the inputs fall inside the attractor.

These results together suggest that gnODEs might be flexible enough to learn more general manifold geometries, provided they are trained on an appropriate synthetic task. Furthermore, the geometry of the low-dimensional representations found by gnODEs can directly inform their generalization capacities, thanks to their enhanced interpretability.

### Acknowledgments

We would like to thank Patrick Kidger for helpful discussions. TDK would also like to thank Carlos Brody for his encouragement and support. This work was supported by a C.V. Starr Fellowship and a CPBF Fellowship (NSF PHY-1734030 to KK), a grant from the Simons Foundation (891851 to TC), and the Howard Hughes Medical Institute Investigator support to Carlos Brody. TC also acknowledges the support of the Eric and Wendy Schmidt Membership in Biology and the Simons Foundation at the Institute for Advanced Study.

### References

Jeff Bezanson, Alan Edelman, Stefan Karpinski, and Viral B Shah. Julia: A fresh approach to numerical computing. SIAM Review, 59(1):65–98, 2017.

# Extended Abstract Track

Tankut Can and Kamesh Krishnamurthy. Emergence of memory manifolds. arXiv, 2021.

Tankut Can, Kamesh Krishnamurthy, and David J. Schwab. Gating creates slow modes and controls phase-space complexity in GRUs and LSTMs. In Proceedings of The First Mathematical and Scientific Machine Learning Conference, pages 476–511, 2020.

Ricky T. Q. Chen, Yulia Rubanova, Jesse Bettencourt, and David K Duvenaud. Neural Ordinary Differential Equations. Advances in Neural Information Processing Systems, 31: 6571–6583, 2018.

Kyunghyun Cho, Bart van Merrienboer, Caglar Gulcehre, Dzmitry Bahdanau, Fethi Bougares, Holger Schwenk, and Yoshua Bengio. Learning phrase representations using rnn encoder-decoder for statistical machine translation, 2014.

Emilien Dupont, Arnaud Doucet, and Yee Whye Teh. Augmented Neural ODEs. In Advances in Neural Information Processing Systems, volume 32, 2019.

Jeffrey L. Elman. Finding structure in time. Cognitive Science, 14(2):179–211, 1990.

Chris Finlay, Joern-Henrik Jacobsen, Levon Nurbekyan, and Adam Oberman. How to train your neural ODE: the world of Jacobian and kinetic regularization. In Proceedings of the 37th International Conference on Machine Learning, volume 119, pages 3154–3164, 2020.

Amir Gholami, Kurt Keutzer, and George Biros. ANODE: Unconditionally Accurate Memory-Efficient Gradients for Neural ODEs. arXiv, 2019.

Arnab Ghosh, Harkirat Behl, Emilien Dupont, Philip Torr, and Vinay Namboodiri. STEER : Simple Temporal Regularization for Neural ODE. In Advances in Neural Information Processing Systems, volume 33, pages 14831–14843, 2020.

Xavier Glorot and Yoshua Bengio. Understanding the difficulty of training deep feedforward neural networks. In Proceedings of the Thirteenth International Conference on Artificial Intelligence and Statistics, volume 9, pages 249–256, 2010.

S. Hochreiter and J. Schmidhuber. Long short-term memory. Neural Computation, 9(8): 1735–1780, 1997.

Mike Innes. Flux: Elegant machine learning with julia. Journal of Open Source Software, 3 (25):602, 2018.

Rafal Jozefowicz, Wojciech Zaremba, and Ilya Sutskever. An empirical exploration of recurrent network architectures. In International conference on machine learning, pages 2342–2350. PMLR, 2015.

Jacob Kelly, Jesse Bettencourt, Matthew James Johnson, and David Duvenaud. Learning differential equations that are easy to solve. In Neural Information Processing Systems, 2020.

Patrick Kidger. On Neural Differential Equations. PhD thesis, Oxford, February 2022. arXiv: 2202.02435.

# Extended Abstract Track

Patrick Kidger, James Morrill, James Foster, and Terry Lyons. Neural Controlled Differential Equations for Irregular Time Series. Advances in Neural Information Processing Systems, 2020.

Timothy Doyeon Kim, Thomas Zhihao Luo, Jonathan W. Pillow, and Carlos D. Brody. Inferring latent dynamics underlying neural population activity via neural differential equations. Proceedings of the 38th International Conference on Machine Learning, 2021.

Kamesh Krishnamurthy, Tankut Can, and David J. Schwab. Theory of gating in recurrent neural networks. Phys. Rev. X, 12:011011, 2022.

Ilya Loshchilov and Frank Hutter. Decoupled weight decay regularization. In International Conference on Learning Representations, 2019.

Niru Maheswaranathan, Alex Williams, Matthew Golub, Surya Ganguli, and David Sussillo. Reverse engineering recurrent networks for sentiment classification reveals line attractor dynamics. Advances in neural information processing systems, 32, 2019.

Warren S. McCulloch and Walter Pitts. A logical calculus of the ideas immanent in nervous activity. The bullletin of mathematical biophysics, 5(4):115–133, 1943.

Patrick Kofod Mogensen and Asbjørn Nilsen Riseth. Optim: A mathematical optimization package for julia. Journal of Open Source Software, 3(24), 2018.

James Morrill, Patrick Kidger, Lingyi Yang, and Terry Lyons. Neural controlled differential equations for online prediction tasks. arXiv, 2021.

Derek Onken and Lars Ruthotto. Discretize-optimize vs. optimize-discretize for time-series regression and continuous normalizing flows. arXiv, 2020.

Avik Pal, Yingbo Ma, Viral Shah, and Christopher V Rackauckas. Opening the blackbox: Accelerating neural differential equations by regularizing internal solver heuristics. In Proceedings of the 38th International Conference on Machine Learning, volume 139, pages 8325–8335, 2021.

Razvan Pascanu, Tomas Mikolov, and Yoshua Bengio. On the difficulty of training recurrent neural networks. In International conference on machine learning, pages 1310–1318. PMLR, 2013.

Christopher Rackauckas and Qing Nie. Differentialequations.jl–a performant and feature-rich ecosystem for solving differential equations in julia. Journal of Open Research Software, 5 (1), 2017.

Christopher Rackauckas, Yingbo Ma, Julius Martensen, Collin Warner, Kirill Zubov, Rohit Supekar, Dominic Skinner, and Ali Ramadhan. Universal differential equations for scientific machine learning. arXiv preprint arXiv:2001.04385, 2020.

Mirco Ravanelli, Philemon Brakel, Maurizio Omologo, and Yoshua Bengio. Light gated recurrent units for speech recognition. IEEE Transactions on Emerging Topics in Computational Intelligence, 2(2):92–102, 2018.

T Konstantin Rusch and Siddhartha Mishra. Coupled oscillatory recurrent neural network (cornn): An accurate and (gradient) stable architecture for learning long time dependencies. arXiv preprint arXiv:2010.00951, 2020.

T Konstantin Rusch and Siddhartha Mishra. Unicornn: A recurrent model for learning very long time dependencies. In International Conference on Machine Learning, pages 9168–9178. PMLR, 2021.

T Konstantin Rusch, Siddhartha Mishra, N Benjamin Erichson, and Michael W Mahoney. Long expressive memory for sequence modeling. arXiv preprint arXiv:2110.04744, 2021.

H. Sompolinsky, A. Crisanti, and H. J. Sommers. Chaos in random neural networks. Phys. Rev. Lett., 61:259–262, 1988.

H. Francis Song, Guangyu R. Yang, and Xiao-Jing Wang. Training excitatory-inhibitory recurrent neural networks for cognitive tasks: A simple and flexible framework. PLOS Computational Biology, 12:1–30, 2016.

David Sussillo and L.F. Abbott. Generating coherent patterns of activity from chaotic neural networks. Neuron, 63(4):544–557, 2009.

David Sussillo and Omri Barak. Opening the Black Box: Low-Dimensional Dynamics in High-Dimensional Recurrent Neural Networks. Neural Computation, 25(3):626–649, 2013.

Tim P. Vogels, Kanaka Rajan, and L.F. Abbott. Neural network dynamics. Annual Review of Neuroscience, 28(1):357–376, 2005.

Saurabh Vyas, Matthew D Golub, David Sussillo, and Krishna V Shenoy. Computation through neural population dynamics. Annual Review of Neuroscience, 43:249–275, 2020.

Guangyu Yang, Madhura Joglekar, H. Song, William Newsome, and Xiao-Jing Wang. Task representations in neural networks trained to perform many cognitive tasks. Nature Neuroscience, 22, 2019.

## Appendix A. Model Details

In our experiments, we use activation $\phi = \mathrm{ReLU}$, and tend to utilize Glorot initialization (Glorot and Bengio, 2010), which places the neural ODEs comfortably in the stable regime. We are able to determine the critical initialization for nODEs, but find that the initialization does not have a significant effect on training and validation losses.

We have used $F_\theta(\mathbf{h}, \mathbf{x}) \overset{\text{def}}{=} \mathbf{s}^L$, where

$$\mathbf{s}^L = W^{L-1}\mathbf{s}^{L-1} + \mathbf{b}^{L-1}, \quad \mathbf{s}^{\ell+1} = \phi(W^\ell \mathbf{s}^\ell + \mathbf{b}^\ell), \quad \mathbf{s}^1 = \phi(W^0 \mathbf{h} + U\mathbf{x} + \mathbf{b}^0),, \quad (2)$$

where $\mathbf{s}^\ell \in \mathbb{R}^{N_\ell}$, the trainable weights and biases are resp. $W^\ell \in \mathbb{R}^{N_{\ell+1} \times N_\ell}$, $\mathbf{b}^\ell \in \mathbb{R}^{N_{\ell+1}}$, and $N_0 = N_L = N$ is the phase-space dimension (dimension of $\mathbf{h}$).

A related model in neuroscience takes $\tau\dot{\mathbf{h}} = -\mathbf{h}(t) + W\phi(\mathbf{h}) + U\mathbf{x}(t)$, where $\mathbf{h}$ can be interpreted as the internal voltage of a neuron, and $\phi(\mathbf{h})$ as the output firing rate of the neuron; $W_{ij}$ is the synaptic strength between neuron $j$ and neuron $i$ (Sompolinsky et al., 1988).

## Appendix B. Related Work

The gnODE architecture is closely related to neural controlled differential equations (nCDEs), developed in Morrill et al. (2021); Kidger et al. (2020), which prescribes a principled way to include inputs with nODEs:

$$\tau\dot{\mathbf{h}} = \tilde{F}_\theta(\mathbf{h})\frac{d\mathbf{x}(t)}{dt}. \quad (3)$$

An important distinction between Equation (1) (setting $\sigma = 1$) and Equation (3) (other than the leak term $-\mathbf{h}$) is that the gnODE takes in input $\mathbf{x}(t)$, whereas nCDE uses the *time-derivative* of the input $d\mathbf{x}/dt$. Because the choice of the interpolation scheme used in Equation (3) also determines how the derivative is estimated, which scheme to use becomes critical (Morrill et al., 2021). Equation (1) avoids the complication of calculating the derivative, though it may not be as general as the nCDE (Kidger et al., 2020).

The primary motivation for introducing gating is its robust ability to generate long timescales and to address the exploding and vanishing gradients problem (EVGP) (Hochreiter and Schmidhuber, 1997; Cho et al., 2014; Pascanu et al., 2013; Can et al., 2020; Krishnamurthy et al., 2022). Our work can be viewed as distilling the key elements of gating from GRUs and LSTMs, responsible for long timescales and stable gradients (cf. Krishnamurthy et al. (2022); Can et al. (2020)), and incorporating them in nCDE-inspired models. Future work could incorporate these gating interactions more directly in nCDEs, with their principled dealing of inputs and interpolation schemes.

Notable recent works have used RNNs based on discretized ODEs to deal with the EVGP, and achieve near state-of-the-art performance on various tasks. A RNN based on a system of coupled non-linear oscillators (coRNN) was introduced in Rusch and Mishra (2020), and this was extended to a Hamiltonian system with multiple (learned) time-scales in Rusch and Mishra (2021). In particular, the presence of the learned timescales was important for solving tasks with long timescales (Rusch and Mishra, 2021). In Rusch et al. (2021), the authors introduced a RNN based on gated ODEs – the long expressive memory (LEM) –

Extended Abstract Track

that makes the timescales *adaptive* and effectively deals with the EVGP. The gated ODEs in Rusch et al. (2021) are very close in spirit to the gated ODEs considered in Krishnamurthy et al. (2022), and in fact, are very similar to the latter ODEs with a certain block-structure for the weight matrices. Given the strong inductive bias of the gating for tasks requiring long memory, our work can also be considered as extending the ODEs considered in Rusch et al. (2021) to incorporate more flexible flow-fields as in nODEs and nCDEs.

Finally, in addition to addressing the EVGP, we show in this work that gating introduces a powerful inductive bias for integrator-like behavior, in the form of continuous attractors (see Section 2).

## Appendix C. Experiment Details

### C.1. Code

All of the networks presented in this work (vanilla RNN, mGRU, GRU, nODE and gnODE) are implemented in Julia (Bezanson et al., 2017), utilizing Flux.jl, DifferentialEquations.jl and DiffEqFlux.jl (Innes, 2018; Rackauckas and Nie, 2017; Rackauckas et al., 2020).

### C.2. Choice of Discretization

In our experiments, we choose to discretize our networks (vanilla RNN, mGRU, GRU, nODE and gnODE) using the canonical forward Euler method. While the optimal choice of discretization method may depend on the problem, we find that the simple Euler solver can achieve strong performance while taking less training time than an adaptive solver in our experiments. Often, the number of function evaluations (NFEs) in a nODE can become extremely large during training for adaptive schemes, and several regularization methods have been introduced to reduce NFEs (Kelly et al., 2020; Ghosh et al., 2020; Finlay et al., 2020; Pal et al., 2021). On the other hand, we can control the NFEs explicitly by changing the timestep $\Delta t$ in a fixed-timestep solver, such as the Euler method. While the Euler method does not have guarantees on the growth of error, it may in fact allow representing more functions compared to adaptive methods that provide such guarantees, precisely because of the errors from the discretization (Dupont et al., 2019). We do not lose the benefit of being able to train nODEs on irregularly-sampled time series when we use the Euler solver. For the $n$-bit flip-flop task in Section 2, changing the Euler method (used for presenting results in the main text) to the Tsitouras 5/4 Runge-Kutta method did not make a significant qualitative difference.

### C.3. Choice of Adjoint

For all networks we consider, we backpropagate through the operations of the solver—that is, we use the "discretize-then-optimize" approach, as is standard in training an RNN, instead of using the "optimize-then-discretize" approach used in Chen et al. (2018) to train nODEs. A few studies show that the former produces more accurate gradients than the latter and can yield better performances (Gholami et al., 2019; Onken and Ruthotto, 2020).

Extended Abstract Track

## Appendix D. $N$-Bit Flip-Flop Task

For all versions of the $n$-bit flip-flop task in Section 2 and this Appendix, the total length of each trial was 1s, binned into 10ms bins. Thus each trial had 100 time-bins. The width of each pulse was set to be 20ms. 600 trials were generated total, where 500 trials were used for training and the remaining 100 trials were used for validation.

Networks considered in Section 2 (vanilla RNN, mGRU, GRU, nODE and gnODE) were initialized with Glorot uniform initialization (Glorot and Bengio, 2010) with zero bias. $\tau = 0.01$s in all of the networks. We used AdamW (Loshchilov and Hutter, 2019) for training.

### D.1. Fixed-Amplitude 3-Bit Flip-Flop Task

This is the version of the task that was originally introduced in Sussillo and Barak (2013). For this task, we determined the total number of pulses (summed across $n$ channels) on each trial by sampling a number $k$ from the Poisson distribution with mean 12. We then randomly chose $k$ indices from 1 to 100 without replacement. These $k$ indices were the indices at which the pulses occur. For each of the $k$ indices, we randomly chose which one of the $n$ channels the pulse will occur. Then for the channel where the pulse appears, we chose either $+1$ or $-1$ randomly as the value to be taken by the pulse (Figure 3A).

We trained our networks on 500 trials of this task for 200 epochs. The initial states of the networks were not learned, and were initialized with $\mathbf{h}_0 \sim \mathcal{N}(\mathbf{0}, \mathbf{\Sigma})$, where $\mathbf{\Sigma} = \frac{2}{N+1}I$ was the variance. For vanilla RNN, mGRU, GRU, we varied the phase-space dimension $N = \{6, 12, 18\}$, where for each $N$, we used the learning rate $\eta = 10^{-2}$, decay rate $w = 10^{-1}$ and the batch size $B = 100$. For nODE and gnODE, we similarly varied $N = \{6, 12, 18\}$, and used $\eta = 10^{-3}$, $w = 10^{-1}$ and $B = 100$. For nODE and gnODE, $F_\theta$ had 3 hidden layers with 100 units each layer (i.e., $L = 4$ and $H = 100$). We logged the validation MSE traces of mGRU, GRU and gnODE of $N = 6$ over 200 epochs (or $200 \times (500/100) = 1000$ iterations), and found that mGRU, GRU and gnODE all achieved validation MSEs $< 0.01$ at least at some point over the 200 epochs. Similarly, we logged the validation MSE traces of vanilla RNN and nODE of $N = 6$. These networks reached minimum validation MSEs of 0.033 and 0.016, respectively, over the 200 epochs. All networks reached minimum validation MSEs $< 0.01$ during 200 epochs when $N = \{12, 18\}$ (Figure 3B). For further analyses of the trained networks (e.g., performing PCA over the trajectories taken by the networks, and finding the fixed points of the networks), we used the set of parameters that achieved the minimum validation MSEs over the 200 epochs. All networks, when the reached minimum validation MSE was $< 0.01$, used similar strategies for this task – the networks created 8 stable fixed points to solve the task, where each of the 8 stable fixed points represented each output that the networks should take (Figure 3C for vanilla RNN; other networks not shown). For details on how the fixed points were found, see Section D.3.

### D.2. Variable-Amplitude 3-Bit Flip-Flop Task

We determined when the pulses occur and in what channel the pulses occur in the same way as Section D.1. Then for the channel where the pulse appears, we drew a sample $m$ from $U[-1, 1]$ and let $m$ be the value to be taken by the pulse (Figure 1A in main text).

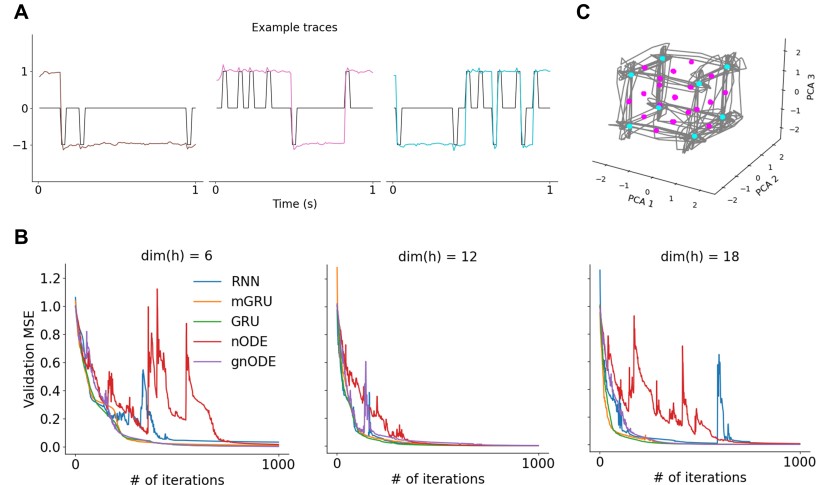

Figure 3: Networks performing the original fixed-amplitude 3-bit flip-flop task Sussillo and
Barak (2013). (A) An example validation trial with inputs in each channel shown
in black, and the trained vanilla RNN traces maintaining the previous pulse value
shown in colors. (B) Validation loss traces as a function of the number of iterations.
(C) The first 3 principal components of the vanilla RNN ($N = 18$) trajectories and
fixed points. Cyan indicates stable fixed point. Magenta indicates unstable fixed
point.

To ensure fair comparisons across different networks (vanilla RNN, mGRU, GRU, nODE
and gnODE), for each network, we ran $3 \times 3 \times 3 = 27$ different configurations of $(\eta, w, B)$,
where $\eta = \{10^{-4}, 10^{-3}, 10^{-2}\}$, $w = \{10^{-3}, 10^{-2}, 10^{-1}\}$ and $B = \{10, 50, 100\}$. For each
network and each configuration, we trained for 600 epochs, and determined the set of
parameters that gives the minimum validation MSE over the 600 epochs. Each circle in
Figure 1B–C in the main text is the minimum validation MSE achieved over 600 epochs for
a single configuration, with a total of 27 circles for each network. For nODE and gnODE, $F_\theta$
had 3 hidden layers with 100 units each layer (i.e., $L = 4$ and $H = 100$). We let $N$ be either
6 (Figure 1B in the main text) or 100 (Figure 1C in the main text).

### D.3. Fixed-Point Finder

The finder should find some $\mathbf{h}$ which satisfies $\dot{\mathbf{h}} \approx \mathbf{0}$. To find such $\mathbf{h}$, we define some function
$f$ such that $\dot{\mathbf{h}} = f(\mathbf{h})$. In the case of a nODE, for example, $f(\mathbf{h}) = (1/\tau) \cdot \left(-\mathbf{h} + F_{\hat{\theta}}(\mathbf{h}, \mathbf{x}(t))\right)$,
where we assume that $\mathbf{x}(t) = \mathbf{0}$, and $\hat{\theta}$ is the set of trained parameters of the nODE. We
used Newton's method (implemented in Julia's NLsolve.jl package; Mogensen and Riseth
(2018)) to find the root of the nonlinear function $f(\mathbf{h})$. From a starting point, we ran the
method for 100 iterations, and terminated whenever $\|f(\mathbf{h})\| < 0.01$. Choosing what starting
point to use can be important, especially when $N$ is large. Following Sussillo and Barak
(2013), we used points in the trajectories taken by the network in the validation trials as the
starting points of the finder. We detected fixed points by running the finder 10,000 times,

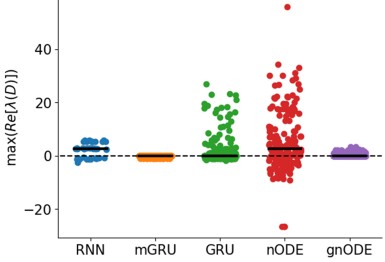

Figure 4: Networks assuming $N = 6$ performing the variable-amplitude 3-bit flip-flop task. Each circle is the maximum real component of the eigenvalues of Jacobian evaluated at a detected fixed point. Bold horizontal lines indicate medians.

each with $10,000$ different starting points. Once we detect some $\mathbf{h}$ that satisfies $\|\dot{\mathbf{h}}\| < 0.01$, we checked whether each element $h_i$ for all $i \in \{1, 2, ..., N\}$ satisfy $2q < h_i < 2r$ to ensure that the detected fixed (or slow) point is not too far from the trajectories taken by the network. Here, $q = \min_{\mathbf{y}}(\min_i(y_i))$ where $\mathbf{y}$ is one of the points in the trajectories taken by the network in the validation trials. Similarly, $r = \max_{\mathbf{y}}(\max_i(y_i))$.

### D.4. Stability of Fixed Points

Figure 1C in the main text suggests that the vanilla RNN ($N = 100$) may be reaching its solution using a combination of marginally-stable and stable fixed points, while the gated networks (mGRU, GRU and gnODE) mostly rely on marginally-stable fixed points. We did a similar analysis for networks assuming $N = 6$ and find similar results (Figure 4). The medians and quartiles of the plotted circles in Figure 1C of the main text and those of Figure 4 are provided in Table 1.

Table 1: We compute $\max(\text{Re}\left[\lambda(\mathcal{D})\right])$, the maximum real component of the eigenvalues of Jacobian at a numerically-detected fixed point. We provide below the medians and quartiles of the distribution of $\max(\text{Re}\left[\lambda(\mathcal{D})\right])$.

|  | N=100 | | N=6 | |
| --- | --- | --- | --- | --- |
| Model | Median | Quartiles | Median | Quartiles |
| RNN | 2.587 | $[-1.969, 6.360]$ | 2.750 | $[-0.768, 4.703]$ |
| mGRU | $-0.472$ | $[-0.473, -0.470]$ | 0.004 | $[0.002, 0.009]$ |
| GRU | $-0.033$ | $[-0.474, 0.070]$ | 0.005 | $[0.001, 0.023]$ |
| nODE | 4.988 | $[-1.627, 9.453]$ | 2.626 | $[-2.030, 13.151]$ |
| gnODE | 0.842 | $[-0.461, 1.698]$ | 0.002 | $[0.002, 0.003]$ |

### D.5. The Family of 2-Bit Flip-Flop Tasks

#### D.5.1. 4 STABLE FIXED POINTS

We determined when the pulses occur and in what channel the pulses occur in the same way as Section D.1, except that now $n = 2$. We trained the gnODE for 200 epochs with 27 different hyperparameter configurations, similar to Section D.2. The initial state of the gnODE was learned – the initial state was assumed to be an affine transformation of the input at the first time-bin. The gnODE's $F_\theta$ had 2 hidden layers with 316 units each layer (i.e., $L = 3$ and $H = 316$). For Figure 2A in the main text, we used the gnODE that reached the lowest validation MSE (3.203e-8) among the 27 different runs.

#### D.5.2. SQUARE ATTRACTOR

We determined when the pulses occur and in what channel the pulses occur in the same way as Section D.2, except that now $n = 2$ (Figure 5A). We trained all networks (vanilla RNN, mGRU, GRU, nODE and gnODE) for 200 epochs, each with 27 different hyperparameter configurations, similar to Section D.2 (Figure 5B). The initial states of the networks were learned – the initial state was assumed to be an affine transformation of the input at the first time-bin. For nODE and gnODE, $F_\theta$ had 2 hidden layers with 316 units each layer (i.e., $L = 3$ and $H = 316$). For Figure 2B in the main text, we used the gnODE that reached the lowest validation MSE (0.008) among the 27 different configurations.

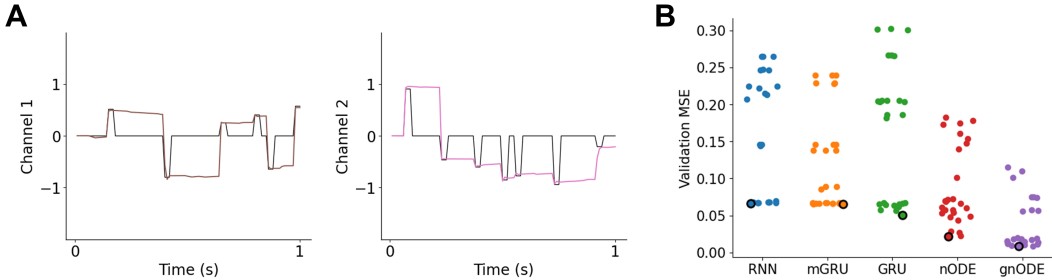

Figure 5: Networks assuming $N = 2$ performing the square 2-bit flip-flop task (Section D.5.2). (A) An example validation trial with inputs in each channel shown in black, and the trained gnODE traces maintaining the previous pulse value shown in colors. (B) For each network, we tried 27 different hyperparameter configurations. Each circle represents the minimum validation MSE achieved during 200 epochs of training. Circles with black edges represent the minimum out of the 27 configurations.

#### D.5.3. RECTANGLE ATTRACTOR

We determined when the pulses occur and in what channel the pulses occur in the same way as Section D.5.2, except that the pulse value for Channel 1 was drawn from $U[-2, 2]$, while the pulse value for Channel 2 was drawn from $U[-1, 1]$ (Figure 6A). For Figure 2C in the main text, we used the gnODE that reached the lowest validation MSE (0.0137) among the

Figure 6: The gnODE assuming $N = 2$ performing the rectangle and the disk 2-bit flip-flop task. (A) An example validation trial on the rectangle task with inputs in each channel shown in black, and the trained gnODE traces maintaining the previous pulse value shown in colors. (B) Same as (A) but for the disk task.

27 different configurations. Architecture used for gnODE for this task was the same as the one used in Section D.5.2.

### D.5.4. DISK ATTRACTOR

We determined the total number of pulses (summed across $n$ channels) on each trial by sampling a number $k$ from the Poisson distribution with mean 6. We then randomly chose $k$ indices from 1 to 100 without replacement. These $k$ indices were the indices at which the pulses occur. For each of the $k$ indices, we drew random samples $C_1$, $C_2$, ..., $C_n$ which satisfy $1 < \sqrt{C_1^2 + C_2^2 + ... + C_n^2} < 2$, where $C_i$ is the pulse value in Channel $i$. Thus the input pulses in $n$ channels appear at the same time. For Figure 2D in the main text, we used the gnODE that reached the lowest validation MSE (1.910e-4) among the 27 different configurations. Architecture used for gnODE for this task was the same as the one used in Section D.5.2. Notice in Figure 2D, we see a disk attractor with radius roughly of 2. We do not see a hole between radius 0 and 1 because crossing this region may be the fastest way from one state to another, and we did not explicitly penalize the network for crossing this region. Consistent with the flow field, we find that, even though the gnODE has not seen any inputs with pulse values satisfying $0 < \sqrt{C_1^2 + C_2^2} < 1$ during training, when it is given inputs with pulse values satisfying $0.5 < \sqrt{C_1^2 + C_2^2} < 1$, it generalizes well (MSE = 0.005; see Appendix D.5 for details). However, when it is given inputs with pulse values that are small (i.e., $0 < \sqrt{C_1^2 + C_2^2} < 0.5$), it does tend to mistake them as having no input at all, resulting in worse performance (MSE = 0.028). When the gnODE was given inputs with pulse values $2 < \sqrt{C_1^2 + C_2^2} < 4$, it did not generalize (MSE = 0.490).

### D.5.5. RING ATTRACTOR

We determined when the pulses occur in a way similar to Section D.5.5. However, $k$ was drawn from the Poisson distribution with mean 12, not 6. Also, the constraint that $C_1$, $C_2$, ..., $C_n$ had to satisfy was $\sqrt{C_1^2 + C_2^2 + ... + C_n^2} = 2$.

We trained the gnODE for 200 epochs with 27 different hyperparameter configurations on this task. Architecture used for gnODE for this task was the same as the one used in Section D.5.2. Figure 7 shows the flow fields of gnODE trained on this task. Figure 7A shows the flow field of gnODE when we let the initial state that the network should take (in

the output space) be $(0,0)$ (validation MSE: 8.345e-5). We see a structure that is similar to what we see when we train the gnODE on the disk 2-bit flip-flop task (Section D.5.4). When we instead let the initial state that the network should take be $(2,0)$, we see a more ring-like structure near $(2,0)$ (Figure 7B; validation MSE: 0.007). We also tried changing the initial state of gnODE from $(0,0)$ to $(1.5,0)$ for the gnODE trained on the disk 2-bit flip-flop task (Section D.5.4), but did not see a significant qualitative difference.

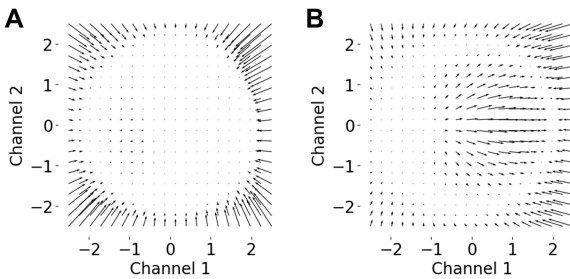

Figure 7: Flow fields of gnODE with $N = 2$ performing the ring 2-bit flip-flop task. (A) The initial state of gnODE is set to be $(0,0)$. (B) The initial state of gnODE is set to be $(2,0)$.

