# OpenReview forum: "Learning and Shaping Manifold Attractors for Computation in Gated Neural ODEs"
_NeurIPS.cc/2022/Workshop/NeurReps — NeurReps 2022 Poster_

### Official Review · Reviewer_87cq · 2022-10-10
**Useful and important model contribution, but multiple technical concerns may hinder recognition by community.**

**Confidence:** 5
**Soundness:** 3
**Presentation:** 4
**Contribution:** 4
**Overall Rating:** 7

**Summary:**

The authors combine the ideas of gating with neural ODEs and find that they can learn plane-shaped attractors in an efficient, interpretable manner with a small latent space. They extend the n-bit flip flop task from Barak and Sussillo (2013) to continuous inputs to demonstrate this. On this task, they show that their model, the gnODE outperforms RNNs, GRUs, and standard neural ODEs when the hidden size of the RNN is constrained to be low dimensional. They visualize the flow field of a 2-dimensional gnODE trained on the 2-bit variable amplitude flip flop and show that the model learns an interpretable disk/square/rectangle attractor, depending on the structure of the input.

**Questions:**

In figure 1c (bottom), I was confused how the RNN, nODE, and gnODE learned so many fixed points with linearized eigenvalues of magnitude >> 1. Wouldn't this make the networks unstable? It would be useful to clarify what these points are, or remove them if they exist in areas of the activity space that are never reached by the networks. Alternatively, if they exist because the continuous attractors are in actuality discretized, it would be helpful to know if these unstable fixed points are in between the fixed points in this attractor for smooth adjustment after input.

How biologically plausible is this model? One of the reasons why RNNs tend to be the model of choice in contemporary computational neuroscience (like what Barak and Sussillo used) is that it is agnostic about neural computations like gating. However, I think it is plausible with neuromodulation, and I would like to see the authors discuss this in the paper. I feel that this would strengthen the paper's contribution.

In the 2-d attractor demonstration, I am curious to know whether the gate outputs 0 or the other side of the multiplicative interaction outputs 0 (-h + F(x,h)). In theory, if it were the "vanilla" side that dropped to zero, the gate might not even matter, right?

**Limitations:**

I am not aware of any important limitations at this time. Perhaps that authors could sweep more architectures (initializations, learning rates, MLP sizes, input channel number) and compare models more thoroughly.

**Recommended Decision:**

3: Accept

**Relevance:**

4: Highly relevant

**Strengths And Weaknesses:**

The model design is innovative, capturing two very useful ideas in deep learning that haven't seen much light in computational neuroscience. The second figure's flow fields were especially striking and demonstrate the interpretability of proposed model. The paper makes a smart extension of an existing task for their design. The description of the model and the task was clear, and the analysis was well described. The Appendix helped answer my clarifying questions as far as model implementation. This model is certainly of interest to the community for its ability to learn interpretable latent geometries. The authors make a strong claim for both gating and neural ode's power and relevance for neural modeling.

I would appreciate it as well if the figures were made larger in the final version.

However, I felt that many of the technical claims need to be qualified further as some could border on potentially misleading the audience.

First, in comparing the learning abilities of the different RNNs, the authors do not include LSTMs for comparison, which is typically considered to be the leading standard for recurrent networks. I think that if you are comparing performances, you must include this model.

Second, the authors compare the models' performances at fixed hidden sizes, but do not account for the fact that the model complexity is significantly greater for the gnODE. In the appendix they explain that their gnODE function $F_\theta(h,x)$ is parameterized as a 4-layer MLP with 100 units in each of the hidden layers, whereas for the RNNs and GRU, this is a simple linear-nonlinear operation $\phi(Wh+Ux(t)+b)$. In the N=6 case, the RNN has (W = 36, U = 18, b = 6) 60 parameters, and the GRU has (2*O(RNN) = 120 parameters, while the gnODE has (MLP: 600+10,000*3) = 30,600 parameters at least. I feel that this scale of difference in  model complexity should be accounted for or mentioned in the paper.

Finally, I am uncertain as far as the usefulness of the ODE in this paper. One of the benefits of using a nODE is that the model can adapt its step size for increased precision, but the authors mention in the appendix that they use Euler integration at a fixed step size. In this case, the model effectively reduces to a GRU with a smaller step size. Thus, I am not sure if the model's power comes from it being an ODE versus this model having more precision in its updates and more expressivity due to overparameterization. This problem should be addressed to argue for the necessity of the ODE (in more complex tasks, I can understand it).

**Submission Track:**

Extended Abstract (4 Page)

---

> ### Author Response · Authors · 2022-11-02
> **Response to Reviewer 87cq**
>
> We thank the reviewer for their careful reading of our abstract and for the constructive feedback. We address some of their concerns below:
>
> >First, in comparing the learning abilities of the different RNNs, the authors do not include LSTMs for comparison, which is typically considered to be the leading standard for recurrent networks. I think that if you are comparing performances, you must include this model.
>
> As we were not entirely focused on comparing performances in this abstract, we did not include LSTMs. In the full paper we are preparing, we include results from a recent, near state-of-the-art RNN model, long expressive memory (LEM) [1].
>
> >Second, the authors compare the models' performances at fixed hidden sizes, but do not account for the fact that the model complexity is significantly greater for the gnODE.
>
> We thank the reviewer for this point. This is an important point that we should mention in the abstract (and now included in the new version). The number of parameters for a low-dimensional GRU is much lower than that of a low-dimensional gnODE. For a typical GRU to achieve overparametrization, we would need to increase its dimensionality. In the other experiments we are conducting in the full paper, we show that even when the numbers of parameters are similar, gnODE can perform similarly or even better than a GRU.
>
> >One of the benefits of using a nODE is that the model can adapt its step size for increased precision, but the authors mention in the appendix that they use Euler integration at a fixed step size.
>
> We thank the reviewer for bringing this up. As the reviewer correctly points out, we used the forward Euler method in the experiments for all network architectures. Because we were less focused on what the effects of discretizations are on performance and more on the effects of network architecture, we decided to go with the simplest one that gives fast training. As we discuss in the Appendix, we found that fixed step size dramatically decreased training time. Depending on different datasets, different discretizations may or may not increase performance of a model. Adaptive schemes are thought to be helpful [2] and can be used not only for nODEs, but also for GRUs [3]. However, as discussed in the Appendix, for the flip flop task, using an adaptive scheme (same across all network architectures) did not make a significant qualitative difference.
>
> >In figure 1c (bottom), I was confused how the RNN, nODE, and gnODE learned so many fixed points with linearized eigenvalues of magnitude >> 1. Wouldn't this make the networks unstable? It would be useful to clarify what these points are, or remove them if they exist in areas of the activity space that are never reached by the networks. Alternatively, if they exist because the continuous attractors are in actuality discretized, it would be helpful to know if these unstable fixed points are in between the fixed points in this attractor for smooth adjustment after input.
>
> We thank the reviewer for bringing up this important point. For Figure 1C, we plotted the spectral abscissa of the numerical Jacobian obtained from fixed points that are reasonably within the space covered by the latent trajectories (see Appendix D.3 for details). However, following the reviewer's suggestion, we explored a different criterion to identify fixed points that are near the latent trajectories -- whenever the identified fixed point is less than 1 in Euclidean distance from any of the points actually traversed by the networks, we include the fixed point in the plot. Even for this criterion, we still saw a result similar to what was presented in Figure 1C. We further projected the 100-dimensional fixed points to the 3-dimensional PC space and found that the unstable fixed points are scattered around the stable fixed points, suggesting that the unstable fixed points may be facilitating the network to fall into one of the stable or marginally stable fixed points.

---

> > ### Author Response · Authors · 2022-11-02
> > **Response to Reviewer 87cq (continued)**
> >
> > >How biologically plausible is this model? One of the reasons why RNNs tend to be the model of choice in contemporary computational neuroscience (like what Barak and Sussillo used) is that it is agnostic about neural computations like gating. However, I think it is plausible with neuromodulation, and I would like to see the authors discuss this in the paper. I feel that this would strengthen the paper's contribution.
> >
> > We thank the reviewer for the suggestion. We agree that one interesting direction to follow up on this work would be to investigate whether and how a gnODE can be linked to a more mechanistic model of network computations [4]. While we do not claim in the abstract that each unit in a gnODE can correspond to a biological neuron, gating appears to be a generally observed phenomenon in a biological neural network. For example, Appendix C of [1] describes how a biologically realistic neuron (a Hodgkin-Huxley neuron in their example) implements gating with voltage-gated ion channels. In another example, negative-derivative feedback in an E-I balanced network can be viewed as a form of gating which dynamically changes the time constant [5].
> >
> > >In the 2-d attractor demonstration, I am curious to know whether the gate outputs 0 or the other side of the multiplicative interaction outputs 0 (-h + F(x,h)). In theory, if it were the "vanilla" side that dropped to zero, the gate might not even matter, right?
> >
> > We thank the reviewer for this point. When we evaluated the gate output and $-\bf{h} + F_\theta(\bf{h},\bf{x})$ on several points inside the 2-D attractor, the norm of $-\bf{h} + F_\theta(\bf{h},\bf{x})$ was generally closer to $0$ than the norm of the gate output. Thus, after training, as the reviewer points out, it may be possible for gnODE to do the task without the gate. However, this does not mean that gates do not help with training/performance. For the points evaluated, we found that each component of the gate had values $\sim 0.1$, and this helps $-\bf{h} + F_\theta(\bf{h},\bf{x})$ become even closer to zero.
> >
> > Relatedly, we also find (not shown in the abstract) that making the gating function be more flexible (e.g., by making it another feedforward neural network) does help with performance in more complicated tasks.
> >
> >
> >
> > [1] Long Expressive Memory for Sequence Modeling. Rusch et al., ICLR 2022.
> >
> > [2] Neural Ordinary Differential Equations. Chen et al., NeurIPS, 2018.
> >
> > [3] Gated Recurrent Units Viewed Through the Lens of Continuous Time Dynamical Systems. Jordan et al., Frontiers in Computational Neuroscience, 2021.
> >
> > [4] Dynamics on the manifold: Identifying computational dynamical activity from neural population recordings. Duncker and Sahani, Current Opinion on Neurobiology, 2021.
> >
> > [5] Balanced cortical microcircuitry for maintaining information in working memory. Lim and Goldman, Nature Neuroscience, 2013.

---

### Official Review · Reviewer_NiS8 · 2022-10-13
**Interesting NN architecture for developing low-dimensional phase space**

**Confidence:** 4
**Soundness:** 3
**Presentation:** 3
**Contribution:** 3
**Overall Rating:** 7

**Summary:**

Authors examined gated neural ODEs that add gated component to neural ODE and characterized the low-dimensional dynamics of trained networks in 3-bit memory task. They showed that gnODE shows particularly good performance compared to other network architectures when the phase-space is low.

**Questions:**

* What aspects of the gnODE makes it possible to learn low-dimensional phase-space better than other architectures?


**Limitations:**

As the authors may be aware, the main claim (gnODE captures low-dimensional representation) is demonstrated only in a toy task. It'd be good to demonstrate similar findings in more challenging tasks.

**Recommended Decision:**

3: Accept

**Relevance:**

3: Solid fit

**Strengths And Weaknesses:**

## Originality
The network architecture is a variant of existing models (nCDF of Morrill et al 2021 or GRU of Cho et al 2014). The novel aspect of the proposed architecture is that it can learn the low-dimensional phase-space representation of tasks.

## Quality
Although the tasks discussed in the manuscript is relatively simple, it demonstrates the proof-of-concept of how gnODE captures the low-dimensional representation.

## Clarity
The manuscript is overall well written. However, it is unclear as to why the proposed gnODE is able to do well when the phase-space is low-dimensional. What aspects of the network makes this possible?

## Significance
This study proposes a potentially useful network architecture for learning low-dimensional representation.


**Submission Track:**

Extended Abstract (4 Page)

---

> ### Author Response · Authors · 2022-11-02
> **Response to Reviewer NiS8**
>
> We thank the reviewer for their constructive feedback of our abstract. We address their concerns below:
>
> >However, it is unclear as to why the proposed gnODE is able to do well when the phase-space is low-dimensional. What aspects of the network makes this possible?
>
> We thank the reviewer for bringing this up. We are currently preparing a full paper that defines the notion of expressivity and explores why nODEs, particularly gnODEs, can learn dynamics in a lower-dimensional phase-space compared to vanilla RNNs, mGRUs and GRUs. The intuition for why is as follows. Overparametrization in a neural network is often critical to its performance, and when RNNs/GRUs are used to model complex dynamics present in real data, this overparametrization typically comes from increasing the RNN phase-space dimension. On the other hand, with nODEs/gnODEs, we may achieve overparametrization by, for example, increasing the depth or width of the hidden layers of the FNN parametrizing the velocity vector field, without expanding the phase-space dimension. This allows low-dimensional nODEs/gnODEs to model complex dynamical flow fields without losing their expressivity.
>
> In addition, we argue that adding gating gives an edge to gnODEs by improving trainability, and in some settings (e.g., the variable-amplitude $n$-bit flip-flop task), supplying a superior inductive bias.
>
> >It'd be good to demonstrate similar findings in more challenging tasks.
>
> We are currently preparing a full paper exploring how different network architectures perform on complex real-world tasks. Our results (not shown in this abstract) suggest that gnODEs can generally perform similarly to GRUs with lower phase-space dimensions.

---

### Official Review · Reviewer_MRQk · 2022-10-15
**The benefits of gating**

**Confidence:** 4
**Soundness:** 3
**Presentation:** 3
**Contribution:** 2
**Overall Rating:** 6

**Summary:**

Inspired by the observation that dynamical systems trained to solve complicated tasks tend to learn low-dimensional dynamics tailored for the task, the authors propose a new parametrization called gated neural ODE. The latent state develops according to a recurrent linear system that gates (multiplicatively) a deep neural ODE. The premise is that this scheme is biased toward learning low-dimensional latent dynamics.

**Questions:**

Have the authors experimented with finding the minimal latent dimension for solving a task?

Can one expect a monotonic improvement in performance (more neurons, better fit), or is there an optimal dimension below which the dynamics are not expressive enough and above which there are too many degrees of freedom to find the “correct” solution?

**Limitations:**

The authors have adequately addressed the limitations of their work.

**Recommended Decision:**

3: Accept

**Relevance:**

3: Solid fit

**Strengths And Weaknesses:**

Strengths:
 * The proposed system is simpler than existing gated architectures, with only N additional units on top of the input dimension. It demonstrated that it still solves tasks where gating is required.
 * The idea to add a minimal amount of gating to a deep neural ODE is original, AFAIK.
 * The proposed system simplifies the interpretation of the system's dynamics due to the low dimensional dynamics.
 * The submission seems technically sound and is clearly written. The claims regarding interpretation are demonstrated in certain simple cases.

Weaknesses:
 * The benefits of this approach are demonstrated on a few tasks, but the main one, where the method is very successful, is modified relative to previous literature (arguably in a way which is favourable for the method in question).
 * The significance of the approach may be limited because there are numerous different parametrizations of ODEs (as the authors review so well in the appendix), and all can be trained to solve similar tasks. It is not fully convincing that the one proposed here is either superior in terms of performance (e.g., with an equal number of units) or in terms of interpretability with respect to a well-known task.

**Submission Track:**

Extended Abstract (4 Page)

---

> ### Author Response · Authors · 2022-11-02
> **Response to Reviewer MRQk**
>
> We thank the reviewer for their constructive feedback of our abstract. We list our response to each of the reviewer's concerns below:
>
> >The significance of the approach may be limited because there are numerous different parametrizations of ODEs (as the authors review so well in the appendix), and all can be trained to solve similar tasks. It is not fully convincing that the one proposed here is either superior in terms of performance (e.g., with an equal number of units) or in terms of interpretability with respect to a well-known task.
>
> We thank the reviewer for bringing up this concern. As the reviewer points out, most of the architectures explored in this abstract were able to solve the flip flop tasks with appropriate hyperparameter-tuning. It is not clear only from the results of the flip flop tasks whether one architecture generally gives superior performance to the others. Furthermore, while the flow fields of the low-dimensional gnODEs in the flip flop tasks can be plotted explicitly and thus are more interpretable than other architectures, the abstract does not mention whether gnODEs can perform tasks in lower dimensions than other architectures for different tasks. We are currently preparing a full paper exploring this aspect, which we hope will address this concern. Our analyses (not shown in the abstract) suggest that gnODEs in general are capable of learning other tasks in lower dimensions, while performing competitively with mGRUs and GRUs.
>
>
> >Have the authors experimented with finding the minimal latent dimension for solving a task?
>
> We thank the reviewer for this question. For the variable-amplitude $n$-bit flip flop task, in principle, latent dimension of $n$ is sufficient to be able to solve the task. The gnODE is capable of learning $2$-bit and $3$-bit flip flop tasks with $2$ and $3$ latent dimensions, repectively. However, we have not done experiments to see if the gnODE can learn the variable-amplitude $3$-bit flip flop task with $1$ or $2$ latent dimensions. In our experiments, the vanilla RNNs, mGRUs and GRUs were not capable of learning the $2$-bit or $3$-bit flip flop tasks using $3$ or less latent dimensions, and required higher latent dimensions.
>
> >Can one expect a monotonic improvement in performance (more neurons, better fit), or is there an optimal dimension below which the dynamics are not expressive enough and above which there are too many degrees of freedom to find the "correct" solution?
>
> We again thank the reviewer for bringing up this important point. In the full paper we are preparing, we formally define the notion of "expressivity" and ask the question of how the topology of the network (e.g., the number of units in a layer or the number of layers) influences the expressivity of the network in a way that is agnostic to a particular task. While we have not done experiments on the variable-amplitude $n$-bit flip flop tasks to identify the "optimal" dimension, we would expect that the "optimal" dimension would be closer to $n$ for gnODEs while it would be higher for other architectures. In principle, we would expect the networks' expressivity to increase monotonically as we increase the latent dimensions, resulting in a saturation in performance after some critical latent dimension. However, in practice, it may well be that there will be some "optimal" peak that is dependent on both the structure and the number of samples in the task/dataset.

---

### Decision · Program_Chairs · 2022-10-21

Accept (Poster)